# Evaluating the Impact of Newborn Screening for Cystic Fibrosis in Portugal: A Decade of Insights and Outcomes

**DOI:** 10.3390/ijns11030069

**Published:** 2025-08-27

**Authors:** Bernardo Camacho, Luísa Pereira, Raquel Bragança, Susana Castanhinha, Raquel Penteado, Teresa R. Silva, Pedro Miragaia, Sónia Silva, Ana L. Cardoso, Telma Barbosa, Cristina Freitas, Juan Gonçalves, Ana Marcão, Laura Vilarinho, Celeste Barreto, Carolina Constant

**Affiliations:** 1Department of Paediatrics, Hospital Dr. Nélio Mendonça, 9000-177 Funchal, Portugal; bernardoncamacho@gmail.com (B.C.);; 2Cystic Fibrosis Reference Centre, Department of Paediatrics, Hospital Santa Maria, Unidade Local de Saúde Santa Maria, 1649-035 Lisbon, Portugal; 3Cystic Fibrosis Reference Centre, Department of Paediatrics, Hospital Dona Estefânia, Unidade Local de Saúde de São José, 1169-045 Lisbon, Portugal; 4Cystic Fibrosis Reference Centre, Department of Paediatrics, Hospital Pediátrico, Unidade Local de Saúde de Coimbra, 3000-602 Coimbra, Portugal; 5Department of Paediatrics, Hospital de São João, Unidade Local de Saúde São João, 4200-319 Porto, Portugal; 6Cystic Fibrosis Reference Centre, Department of Paediatrics, Hospital de São João, Unidade Local de Saúde São João, 4200-319 Porto, Portugal; 7Cystic Fibrosis Reference Centre, Department of Paediatrics, Centro Materno Infantil do Norte, Unidade Local de Saúde Santo António, 4050-651 Porto, Portugal; 8Department of Paediatrics, Hospital Divino Espírito Santo, 9500-370 Ponta Delgada, Portugal; 9Neonatal Screening Unit, National Institute of Health Dr Ricardo Jorge, 1649-016 Porto, Portugal

**Keywords:** newborn screening, cystic fibrosis, sensitivity, false negatives, positive predictive value

## Abstract

The implementation of newborn screening (NBS) has revolutionized the diagnostic landscape of cystic fibrosis (CF). In Portugal, NBS was initiated in October 2013 through a pilot study and was subsequently fully integrated into a nationwide program by December 2018. Infants with positive screening results are referred to a specialized CF reference center for diagnostic confirmation, employing Sweat Chloride Testing (SCT) and genetic testing for *CFTR* variants. We aimed to analyze infants with a positive CF screening and determine the false positive and false negative rates, as well as to calculate the positive predictive value and sensitivity of our NBS program. A retrospective nationwide analysis was conducted on infants with a positive NBS for CF between October 2013 and February 2023. Two hundred and forty infants were referred from the NBS program; 74 (30.8%) were confirmed to have CF through SCT and genetic testing. Sensitivity was 93.2%, and the positive predictive value (PPV) was 30.8%. In addition, 48.5% were homozygous for F508del variants, and 87.8% had at least one F508del variant. Guidelines set forth by the European Cystic Fibrosis Society advise NBS programs to achieve a minimum PPV of 30% and a minimum sensitivity of 95%. Our report demonstrated good compliance with these recommendations.

## 1. Introduction

Cystic fibrosis (CF) is a severe, life-shortening genetic disorder that affects an estimated 100,000 individuals worldwide. Significant improvements in patient outcomes and survival rates have been achieved over the past few decades, largely due to advancements in therapeutic interventions, the adoption of multidisciplinary care, and the implementation of early diagnostic strategies [1].

Newborn screening (NBS) programs for cystic fibrosis are a well-established public health strategy that offers numerous benefits: early diagnosis, followed by timely treatment, leads to reduced morbidity and improvements in lung function, cognitive development, nutrition, and overall growth [2]. It also decreases hospitalization rates and extends life expectancy while enabling families to receive early genetic counseling [3].

The inception of NBS programs for CF dates back nearly 40 years [4]. In our country, a pilot NBS program for CF was initiated in October 2013, which included the screening of 255,000 newborns, and by December 2018, it was fully integrated into the national newborn screening panel. Infants with abnormal screening results are subsequently referred to specialized centers for confirmatory testing, including Sweat Chloride Testing (SCT) and, when appropriate, genetic analysis [5,6,7].

The primary objective of this study was to assess the outcomes of infants referred to CF centers following a positive screening result. Specifically, we aimed to calculate the rates of false positives and false negatives, as well as determine the positive predictive value and sensitivity of the screening program within the Portuguese population.

## 2. Materials and Methods

This was a retrospective, longitudinal, multi-center, and descriptive study, where we analyzed the data on children with NBS positive for CF between October 2013 and February 2023, in all 5 pediatric CF reference centers in Portugal. Among these children, we further analyzed those with a confirmed diagnosis of CF.

For infants with an intermediate or abnormal SCT and/or CF-related variants detected through genetic testing, confirming the CF diagnosis, the following variables were recorded: birth weight, age at diagnosis, age at first evaluation in a CF center, symptoms at presentation, sweat chloride levels, genetic analysis, age at first sputum sample collection, and presence of bacteria in first sputum sample.

The NBS program for CF in Portugal is integrated into the national program for neonatal screening, and although participation is not mandatory, it achieves a coverage rate of 99% to 100% of all live births. The screening process begins with a dried blood spot sample collected between the 3rd and 6th day of life to measure the level of immunoreactive trypsinogen (IRT). Up until February 2023, if the IRT level exceeded 65 ng/mL, the pancreatitis-associated protein (PAP) level was measured. If the PAP level was greater than 1.8 ng/mL, a second dried blood sample was collected between the 3rd and 4th week of life. If the IRT level in the second sample was above 50 ng/mL, the infant was classified as CF-NBS-positive and referred to a CF center for confirmatory testing with an SCT.

From February 2023 onwards, when the IRT level exceeded 65 ng/mL and PAP was over 1.8 ng/mL, genetic testing was performed using the Devyser *CFTR* 68 kit (Devyser, Stockholm, Sweden). If no *CFTR* variants were detected and the IRT level was below 150 ng/mL, the screening was considered negative. However, if the IRT level was 150 ng/mL or above, without any detected variants, a second sample was collected for further IRT evaluation. If this subsequent IRT level fell below 100 ng/mL, the result was negative; if above 100 ng/mL, the screening remained positive despite the absence of variants. In cases where any *CFTR* variants were identified, the screening was deemed positive. After January 2024, the cases with only one variant identified were considered NBS-positives if the IRT exceeded 35 ng/mL in a second sample. The CF newborn screening (NBS) algorithm in Portugal has evolved in response to accumulating data and international evidence on test performance and clinical outcomes. Initially, the IRT/PAP/IRT protocol was implemented to balance sensitivity with a relatively low false positive rate. However, despite its cost-effectiveness and simplicity, this algorithm was associated with suboptimal sensitivity and the logistical burden of recalling a second sample for a considerable proportion of newborns, potentially delaying diagnosis and increasing parental anxiety.

CF is diagnosed when, in an infant with suggestive symptoms, a family history of CF, or a positive NBS for CF, one SCT yields results of 60 mmol/L or higher and/or two pathogenic CF variants are identified through DNA analysis [8]. Infants who have a positive NBS but present with SCT levels below 30 mmol/L, along with 2 CF-causing *CFTR* variants, at least 1 of which has unclear phenotypic consequences, or those with SCT levels between 30 and 59 mmol/L and 0 or 1 CF-causing *CFTR* variants, are classified as CF screen-positive, inconclusive diagnosis (CFSPID) [9].

Confidentiality of patients’ data was maintained throughout this study.

Statistical analysis was conducted using the SPSS 28.0.0 (IBM Corp.) software.

## 3. Results

Between October 2013 and February 2023, a total of approximately 804,000 newborns were screened, resulting in the referral of 240 infants to Portuguese CF reference centers through the NBS program. Among these, 74 (30.8%) were confirmed to have CF through SCT, 6 of whom presented with meconium ileus in the neonatal period. Additionally, during the study period, there were eight cases of CFSPID and eight false negatives (three of whom presented with meconium ileus at birth).

In the population of infants whose diagnosis was confirmed through SCT (Table 1), 39 were female (52.7%), the median age at the first evaluation in a CF center was 33.5 days (IQR 27–40.5), and the average first IRT value was 185.5 ng/mL. Thirty-five (48.5%) infants were homozygous for F508del variants, and 87.8% had at least one F508del variant. The median birth weight was 2943 g (IQR 2615–3263), and the mean sweat chloride level was 87 mmol/L. In addition, 78.3% of the infants had pancreatic insufficiency at presentation, and 74.6% had bacterial growth in sputum samples at the time of the first evaluation at the CF center. The remaining symptoms of these infants at presentation are shown in Table 1.

The overall sensitivity of the screening program in our population was 93.2%, calculated after excluding cases of meconium ileus (MI); specifically, there were 6 patients with MI among the true positives, resulting in 68 remaining cases (74 − 6 = 68). An additional 3 MI cases were reported among the false negatives, yielding a total of 73 relevant cases (i.e., 8 − 3 = 5 FN cases included in the sensitivity calculation; TP = 68; FN = 5), resulting in a sensitivity of 68/73 (93.2%). In accordance with the ECFS Best Practice Guidelines, we also calculated sensitivity after including MI cases, which resulted in 74/82 (90.2%). The positive predictive value (PPV) was 30.8%. The rate of false negatives was 7%, and the false positive rate was 0.21 per 1000 screened newborns.

## 4. Discussion

The European Cystic Fibrosis Society (ECFS) recommends that NBS programs should aim for a minimum PPV of 30% and a minimum sensitivity of 95% [8]. In 2017, Barben et al. showed that countries have very different strategies with regard to the NBS program and that only 62% of the countries met the recommended PPV and 69% met the recommended sensitivity [10]. This shows the importance of reviewing the protocols and assessing the results. In our study, we concluded that the program had a sensitivity of 93.2% in the first 10 years of NBS, which falls just short of the ECFS recommendations. The PPV in our study was 30.8%, which is in line with the recommendations. Our rate of false negatives was 7%, with an overall false positive rate of 0.21 per 1000 screened newborns.

International experience shows that the choice of the CF-NBS algorithm has a major impact on program performance indicators. IRT/IRT algorithms are simple and cost-effective but are often associated with lower sensitivity and the logistical burden of requesting a second dried blood spot, which may delay diagnosis [4,10]. Incorporating a PAP tier (IRT/PAP/IRT or IRT/PAP/DNA) can improve sensitivity and reduce the recall rate, although the balance between sensitivity and false positives depends strongly on the selected cut-offs [5,10]. DNA-based algorithms (IRT/DNA or IRT/PAP/DNA) have demonstrated increased sensitivity, particularly for infants with milder or pancreatic-sufficient phenotypes, but they also result in a higher number of CFSPID and carrier identifications, which can increase clinical uncertainty and the need for counseling [8,9]. For example, Sweden’s IRT/PAP/IRT approach offers good sensitivity (~92%) and reduces false positives compared to simpler IRT-only algorithms, but managing CFSPID remains a challenge [9,10].

There were eight false negative cases in total, three of whom presented with meconium ileus at birth, and the CF diagnosis was not missed in the neonatal period. The remaining cases involved either less common *CFTR* variants, pancreatic sufficiency at the time of diagnosis, or issues related to sample loss and handling (affecting two cases).

The data collected also provide valuable insights into the early presentation and clinical characteristics of infants diagnosed with CF following a positive NBS result. A high proportion (78.3%) of infants were diagnosed with pancreatic insufficiency (PI), a key indicator of disease severity. This percentage is consistent with data showing that CF patients with F508del variants often experience early PI [11]. The high rate of PI underscores the importance of prompt nutritional support to prevent malnutrition and associated complications, particularly as 29.3% of infants exhibited insufficient weight gain at the first evaluation.

The median age for sputum sample collection was slightly over two months, which may reflect the logistics of securing samples in infants. A wider spread in age (IQR) points to variability in clinical follow-up or parental reporting. Given the importance of early detection of pathogens in CF management, streamlining sample collection protocols may be beneficial.

A high rate of bacterial isolation (74.6%) was detected early, which highlights the aggressive nature of CF-related lung disease, even in early infancy, and underscores the need for early respiratory therapies. The early detection of *P. aeruginosa* in 6.7% of infants is also concerning, as this pathogen is associated with more rapid lung function decline and worse long-term outcomes [12]. This finding reinforces the importance of implementing aggressive eradication protocols as soon as *P. aeruginosa* is detected to prevent chronic colonization [13].

The ECFS also recommends that any infant with a diagnosis of CF following a positive NBS should be seen in a CF center by 35 days after birth [8]. Our infants were observed for the first time at 33.5 days of age (median), and 32 newborns (43.2%) were observed at or beyond 35 days of age, while the remaining infants (42, 56.7%) were assessed before reaching 35 days of age, therefore complying with the recommendations.

Timely initiation of treatment following a positive NBS result plays a critical role in influencing early nutritional status and growth outcomes in infants with CF. Recent data from a U.S. registry-based study demonstrated that infants who received their initial CF care at an earlier age (median 10 days) exhibited significantly better weight-for-age and height-for-age z-scores at both the first clinical encounter and at one year of age, compared to those whose care began later (median 47 days) [14]. These findings underscore the importance of minimizing delays between diagnosis and clinical management, as even relatively short differences in timing may have a measurable and lasting impact on growth trajectories during early childhood.

Countries need to reflect critically on their NBS programs so that they can be optimized. This can only be done through rigorous quality monitoring, regular protocol audits, and effective communication among responsible stakeholders. We are confident that the new CF-NBS algorithm will place us on the right path toward improving sensitivity even further while reducing the false positive rate even more and, in turn, minimizing the anxiety these events can cause for patients and families.

## Figures and Tables

**Table 1 IJNS-11-00069-t001:** Clinical features and laboratory findings of infants with a positive screening and a CF diagnosis.

	Positive Screening + CF Diagnosis (n = 74)
Age at first evaluation in a reference center (day), median (IQR)	33.5 (27.0–40.5)
Sweat test (mean, mEq/L)	87
Pancreatic insufficiency, n (%)	58 (78.3)
Genetic analysis	
At least one F508del variant, n (%)	65 (87.8)
Homozygous for F508del variants, n (%)	35 (48.5)
First evaluation	
Birth weight (gr), median (IQR)	2943 (2615–3263)
Steatorrhea, n (%)	27 (36.5)
Insufficient weight gain, n (%)	22 (29.3)
Reluctance to feeding, n (%)	4 (5.3)
Irritability, n (%)	3 (4)
Pulmonary infection, n (%)	7 (9.3)
Microbiology	
Age at first sputum sample collection (day), median (IQR)	64 (42.2–125)
Detection of bacteria in the first sputum sample, n (%)	56 (74.6)
Detection of *Pseudomonas aeruginosa* in the first sputum sample, n (%)	5 (6.67)

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
