# Peer review of "Evaluating the Impact of Newborn Screening for Cystic Fibrosis in Portugal: A Decade of Insights and Outcomes"

_2409-515X, 2025, doi:10.3390/ijns11030069_

Round 1
Reviewer 1 Report
Comments and Suggestions for Authors
Camacho, Constant and colleagues describe the implementation of Cystic Fibrosis (CF) screening in Portugal, which began as a pilot in 2013 and was added to the panel in 2018. The information is of interest to the CF newborn screening community. Including additional details would benefit the manuscript.
Please address the following comments and questions in the manuscript.
- How many babies were screened during this period, and what is the incidence of CF in Portugal? This could be stratified into the pilot and fully integrated periods.
- Does the Newborn Screening Program screen all infants in Portugal, or is there some type of regionalization?
- What was the difference between the pilot study and when CF was fully integrated? For example, were only a subset of infants screened during the pilot? Were screen positive infants handled differently during the pilot period? Please explain the differences.
- What was the rationale for the cutoff changes? For example, to reduce the false positive rate? These data were included in the authors previous work, so this could be broadly noted and referenced.
- Please use recommended gene and variant terminology and format: Use ‘variant’ instead of ‘mutation’, with American College of Medical Genetics and Genomics (ACMG) qualifiers if/where necessary.
- Introduction, lines 55-56: Why are improvements in “cognitive development” referenced for CF?
- Introduction, lines 86-89: Text lists actions if IRT levels are less than or greater than a specified IRT value; e.g., 150ng/ml. What if the IRT equals 150 ng/ml – is this a negative or positive result?
- The citations are not formatted properly at the end of the introduction and elsewhere, including the discussion.
- Infants with 2 CF-causing variants and sweat chloride below the diagnostic range were classified as CFSPID. Does the diagnostic algorithm take CFTR variants that are known to be pathogenic but result in low sweat chloride into account (e.g., 3849+10KbC>T)? What if an infant with symptoms suggestive of CF has 2 CF-causing variants with one known to be associated with lower sweat chloride and sweat chloride of 29 mmol/L? Shouldn’t these individuals also be classified as CF? Please clarify.
- Please list the number of confirmed cases with 2 panel variants, 1 panel variant and no panel variants.
- If possible, it would be of interest to list the other CFTR variants identified with counts for the 82 confirmed cases, either via the NBS panel or at the time of diagnosis.
- Please provide the numerator and denominators for all calculations (sensitivity, %FN, %FP) If there were 74 confirmed cases detected by screening plus 8 false negatives, shouldn’t the sensitivity be 90% (74/82) and the false negative rate be 10% (8/82)? I believe 3 cases that had early diagnosis due to MI were added to the sensitivity estimate. Per lines 104-105, only 74 of the confirmed cases were referred, and so the sensitivity of screening should be based on 74 screen positive infants. I think it would be fair to report that 93% (N=77; or possibly more) had early diagnoses, but sensitivity should be 90%.
- Please elaborate on the “issues related to sample loss and handling”.
- Mean (or median; see next comment) sweat chloride of 87.45 mmol/L implies a degree of precision that isn’t likely to be biologically meaningful. I suggest reporting as 87 mmol/L.
- The age at sweat chloride is listed as the mean value in the text but as the median value in the table; please clarify.
- The age at first evaluation seems relatively late, given the importance of early diagnosis, as the authors point out in the discussion. When are NBS results reported? Are there difficulties getting families into clinic for a sweat chloride test? Based on the information report, it seems like there is likely a 2-week period between reporting and confirming the diagnosis.
- Per the discussion, “The ECFS also recommends that any infant with a diagnosis of CF following a positive NBS should be seen in a CF centre by 35 days after birth.7 Our infants were observed for the first time at 33.5 days of age, therefore complying with the recommendations.” The age reported in the table is a median (or mean), so there are values above and below. How many infants were diagnosed before and after 35 days? It would be helpful to report the number and proportion of infants with diagnosis within (and after) the recommended guidelines.
- Lines 132-143: “pancreatic sufficiency at the time of diagnosis,” is implied as contributing to the cases missed by screening, but how does this influence whether they screened positive? Were the IRT and PAP values low for these cases?
- Minor point: CFTR should be italicized.
Author Response
Comment 1: How many babies were screened during this period, and what is the incidence of CF in Portugal? This could be stratified into the pilot and fully integrated periods.
> Response: 764215 babies were screened during this period. Incidence in Portugal is estimated to be around 1 case per 8000 live births.
Comment 2: Does the Newborn Screening Program screen all infants in Portugal, or is there some type of regionalization?
> Response: Our screening program is nationwide and not mandatory, however, it boasts a coverage of nearly 100% of all newborns.
Comment 3: What was the difference between the pilot study and when CF was fully integrated? For example, were only a subset of infants screened during the pilot? Were screen positive infants handled differently during the pilot period? Please explain the differences.
> Response: The screening program was implemented following the same procedures established during the pilot study. Throughout the pilot phase, all newborns were systematically screened, as continues to be the practice today, and cases with positive screening results were referred to designated Cystic Fibrosis reference centres.
Comment 4: What was the rationale for the cutoff changes? For example, to reduce the false positive rate? These data were included in the authors previous work, so this could be broadly noted and referenced.
> Response: We have added a paragraph regarding this, from line 93 onwards, since another revisor had the same opinion. "The CF newborn screening (NBS) algorithm in Portugal has evolved in response to accumulating data and international evidence on test performance and clinical outcomes. Initially, the IRT/PAP/IRT protocol was implemented to balance sensitivity with a relatively low false-positive rate. However, despite its cost-effectiveness and simplicity, this algorithm was associated with suboptimal sensitivity and the logistical burden of recalling a second sample for a considerable proportion of newborns, potentially delaying diagnosis and increasing parental anxiety".
In addition, to improve diagnostic accuracy and align with European best practices, Portugal adopted an IRT/PAP/DNA algorithm in February 2023. The incorporation of DNA testing—specifically using the Devyser CFTR 68 kit—enhanced sensitivity by identifying newborns with one or two pathogenic variants, thus reducing the number of missed cases. However, the addition of molecular testing also introduced challenges, including a higher rate of detection of carriers and CFSPID (Cystic Fibrosis Screen Positive, Inconclusive Diagnosis) cases, which may lead to uncertainty in clinical follow-up and counseling.
Comment 5: Please use recommended gene and variant terminology and format: Use ‘variant’ instead of ‘mutation’, with American College of Medical Genetics and Genomics (ACMG) qualifiers if/where necessary.
> Response: Have done so.
Comment 6: Introduction, lines 55-56: Why are improvements in “cognitive development” referenced for CF?
> Response: As stated in this article from the CDC (https://pubmed.ncbi.nlm.nih.gov/15483524/), "When CF is diagnosed and treated early, infant's food intake and digestion, growth, cognitive development and lung function can be improved…" I may add this as a reference if you feel it would be more informative.
Comment 7: Introduction, lines 86-89: Text lists actions if IRT levels are less than or greater than a specified IRT value; e.g., 150ng/ml. What if the IRT equals 150 ng/ml – is this a negative or positive result?
> Response: I should have specified that the actions occur if the value is over OR equal in the algorithm, have changed that in the text. (lines 86 and 87)
Comment 8: The citations are not formatted properly at the end of the introduction and elsewhere, including the discussion.
> Response: Have formatted them properly now.
Comment 9: Infants with 2 CF-causing variants and sweat chloride below the diagnostic range were classified as CFSPID. Does the diagnostic algorithm take CFTR variants that are known to be pathogenic but result in low sweat chloride into account (e.g., 3849+10KbC>T)? What if an infant with symptoms suggestive of CF has 2 CF-causing variants with one known to be associated with lower sweat chloride and sweat chloride of 29 mmol/L? Shouldn’t these individuals also be classified as CF? Please clarify.
> Response: In our study, we adhered to the diagnostic guidelines and definitions used at the national level in Portugal during the study period, as implemented across all five CF reference centres. These align closely with the consensus guidelines of the CF Foundation (CFF) and the European CF Society (ECFS).
Specifically:
Infants with two CFTR variants, at least one of which is of unclear or variable clinical consequence (e.g., associated with residual function or variable expressivity), and with sweat chloride values below 60 mmol/L, were classified as CFSPID (CF Screen Positive, Inconclusive Diagnosis), consistent with the ECFS definition.
Variants known to be CF-causing but associated with lower or intermediate sweat chloride levels—such as 3849+10kbC>T— were interpreted in context, taking into account the clinical presentation and the presence of pancreatic or respiratory involvement. However, if the sweat chloride level was below 30 mmol/L and the infant was asymptomatic or only mildly symptomatic, the case was generally classified as CFSPID rather than classic CF.
Regarding the example posed:
“An infant with symptoms suggestive of CF, two CF-causing variants (one associated with lower sweat chloride), and a sweat chloride of 29 mmol/L”
Such a case would typically trigger close clinical follow-up and repeat testing, and in some cases, reclassification to CF may occur if symptoms progress and/or additional biomarkers support disease expression. Indeed, the diagnosis of CFSPID is not static, and periodic re-evaluation is part of the clinical management protocol in Portugal.
Comment 10 and 11: Please list the number of confirmed cases with 2 panel variants, 1 panel variant and no panel variants.
If possible, it would be of interest to list the other CFTR variants identified with counts for the 82 confirmed cases, either via the NBS panel or at the time of diagnosis.
> Response: Thank you for the suggestion. We would be happy to compile the requested list and propose including it as a supplementary table to accompany the manuscript, should the editorial team consider it appropriate.
Comment 12: Please provide the numerator and denominators for all calculations (sensitivity, %FN, %FP) If there were 74 confirmed cases detected by screening plus 8 false negatives, shouldn’t the sensitivity be 90% (74/82) and the false negative rate be 10% (8/82)? I believe 3 cases that had early diagnosis due to MI were added to the sensitivity estimate. Per lines 104-105, only 74 of the confirmed cases were referred, and so the sensitivity of screening should be based on 74 screen positive infants. I think it would be fair to report that 93% (N=77; or possibly more) had early diagnoses, but sensitivity should be 90%.
> Response: The reported sensitivity of 93% was calculated after excluding cases of Meconium ileus (MI) from the denominator. If deemed appropriate, we are happy to clarify this in the manuscript. Specifically, there were 6 patients with MI among the true positives, resulting in 68 remaining cases (74 – 6 = 68). An additional 5 MI cases were reported among the false negatives, yielding a total of 73 relevant cases (68 + 5 = 73), and thus a sensitivity of 68/73 (93%).
Comment 13: Please elaborate on the “issues related to sample loss and handling”.
> Response: Due to the loss of the initial two samples, timely diagnostic results were not available for the children. As a result, sample recollection was delayed beyond the optimal time window, which likely contributed to false-negative outcomes in both instances.
Comment 14: Mean (or median; see next comment) sweat chloride of 87.45 mmol/L implies a degree of precision that isn’t likely to be biologically meaningful. I suggest reporting as 87 mmol/L.
> Response: Have done so.
Comment 15: The age at sweat chloride is listed as the mean value in the text but as the median value in the table; please clarify.
> Response: It is the median, have corrected it in the text.
Comment 16: The age at first evaluation seems relatively late, given the importance of early diagnosis, as the authors point out in the discussion. When are NBS results reported? Are there difficulties getting families into clinic for a sweat chloride test? Based on the information report, it seems like there is likely a 2-week period between reporting and confirming the diagnosis.
> Response: NBS results are typically reported to the reference centres within 2 to 3 weeks. In the majority of cases, there are no significant barriers to promptly referring families to these centres; however, a small number of outliers may delay referral timelines and contribute to increased variability in the data.
Comment 17: Per the discussion, “The ECFS also recommends that any infant with a diagnosis of CF following a positive NBS should be seen in a CF centre by 35 days after birth.7 Our infants were observed for the first time at 33.5 days of age, therefore complying with the recommendations.” The age reported in the table is a median (or mean), so there are values above and below. How many infants were diagnosed before and after 35 days? It would be helpful to report the number and proportion of infants with diagnosis within (and after) the recommended guidelines.
> Response: Added the following sentence to clarify: "Of the cohort, 32 newborns (43%) had their first evaluation at a CF reference centre at or beyond 35 days of age, while the remaining infants (42, 57%) were assessed before reaching 35 days." (lines 163, 164)
Comment 18: Lines 132-143: “pancreatic sufficiency at the time of diagnosis,” is implied as contributing to the cases missed by screening, but how does this influence whether they screened positive? Were the IRT and PAP values low for these cases?
> Response: Some studies have shown that IRT-based screening may miss some CF infants with pancreatic suficiency, due to IRT levels sometimes falling below the set cut-off thresholds (such as https://www.cysticfibrosisjournal.com/article/S1569-1993(16)30025-X/fulltext), also https://pubmed.ncbi.nlm.nih.gov/33072991/, where in Spain’s CF NBS program reported false negatives in PS infants attributed to sub-threshold IRT values.
Comment 19: Minor point: CFTR should be italicized.
> Response: Have done so.
Reviewer 2 Report
Comments and Suggestions for Authors
The study about CF-screening in Portugal over the last 10 years is very interesting as it evaluates the algorithm used in Portugal and provides important information on quality criteria. However, I believe the paper could be substantially improved by adding or improving some points:
In general:
Please ensure that all references are in square brackets e.g., line 96, 99, 123,156.
All percentages should have one number after the decimal point.
Material and Methods:
Maybe it is possible to provide some more information about the CF-screening in Portugal, such as whether all children are screened and whether the screening is mandatory? Is it mandatory to refer to one of the five CF centers for confirmatory diagnostics or are there also other clinics that perform the sweat test?
Changes in the algorithm over time are very interesting. It would be useful to include a figure or schema showing the different algorithms.
Results
It would be interesting to know how many children were born and screened (coverage rate) in the pilot project and after the full implementation beginning December 2018. Some of the given results are unclear for me:
In my opinion, CFSPID can also be “cases”? It should be mentioned if you included these 8 children in your 74 cases and in the calculations or not.
Can you please explain how the 5 false negatives without MI have been found? To calculate the sensitivity, you need to know about all cases in Portugal that were missed by the screening process. Was there a survey for missed cases or do you have a register in Portugal? Additionally, I cannot calculate the sensitivity of 93% with the given numbers. When I calculate the number of cases found in screening (74 or 82 with CFSPID, true positives) divided by the total number of confirmed cases (82 or 90), I get a sensitivity of 90.2% or 91.1%. I also get a different number for the false positives. Therefore, I would suggest explaining and proving all calculations.
Discussion
It would be interesting to know why the CF screening in Portugal was changed twice and discuss at least shortly the advantages and disadvantages of the different algorithms that are used in Portugal and possibly in some other countries and add some literature to these points (e.g., higher sensitivity, high rate of false positives and PPV, necessity of a second dried blood sample, what to do with CFSPID and heterozygotes….).
Additionally, literature should be added to support other points in the discussion, e.g., PI as key indicator (line 138), and patients with F508del experience early PI (line 139), Pseudomonas associated with worse long-term outcomes (line 152) and importance of eradication (line 154), anxiety for families with false positive screening results (line 163)…..
Author Response
Please ensure that all references are in square brackets e.g., line 96, 99, 123,156.
> Response: Have done so.
All percentages should have one number after the decimal point.
> Response: Do you mean that every % should go from 30% to 30.0% ? We do not have many percentages with decimal points.
Material and Methods:
Maybe it is possible to provide some more information about the CF-screening in Portugal, such as whether all children are screened and whether the screening is mandatory? Is it mandatory to refer to one of the five CF centers for confirmatory diagnostics or are there also other clinics that perform the sweat test?
> Response: Although participation in the national newborn screening (NBS) program in Portugal is not mandatory, it achieves a coverage rate of nearly 100% of all live births. Referral of screen-positive cases for cystic fibrosis (CF) to one of the five designated paediatric CF reference centres is mandatory. Additionally, some reference centres operate affiliated satellite units, such as those located in Ponta Delgada (Azores) and Funchal (Madeira), which are coordinated under the Lisbon reference centre.
I have added this sentence in line 77: "and although participation is not mandatory, it achieves a coverage rate of nearly 100% of all live births."
Changes in the algorithm over time are very interesting. It would be useful to include a figure or schema showing the different algorithms.
> Response: I'm afraid I was not able to do so.
Results
It would be interesting to know how many children were born and screened (coverage rate) in the pilot project and after the full implementation beginning December 2018. Some of the given results are unclear for me:
In my opinion, CFSPID can also be “cases”? It should be mentioned if you included these 8 children in your 74 cases and in the calculations or not.
> Response: We did not consider them to be cases and therefore did not include them in the calculations.
Can you please explain how the 5 false negatives without MI have been found? To calculate the sensitivity, you need to know about all cases in Portugal that were missed by the screening process. Was there a survey for missed cases or do you have a register in Portugal? Additionally, I cannot calculate the sensitivity of 93% with the given numbers. When I calculate the number of cases found in screening (74 or 82 with CFSPID, true positives) divided by the total number of confirmed cases (82 or 90), I get a sensitivity of 90.2% or 91.1%. I also get a different number for the false positives. Therefore, I would suggest explaining and proving all calculations.
> Response: We contacted all participating centers to provide case-level data, including any false negative results identified within the same time period. However, the information received on false negatives was limited, and we regret that further data are unlikely to be obtained within a reasonable timeframe.
The reported sensitivity of 93% was calculated after excluding cases of Meconium ileus (MI) from the denominator. If deemed appropriate, we are happy to clarify this in the manuscript. Specifically, there were 6 patients with MI among the true positives, resulting in 68 remaining cases (74 – 6 = 68). An additional 5 MI cases were reported among the false negatives, yielding a total of 73 relevant cases (68 + 5 = 73), and thus a sensitivity of 68/73 (93%).
Discussion
It would be interesting to know why the CF screening in Portugal was changed twice and discuss at least shortly the advantages and disadvantages of the different algorithms that are used in Portugal and possibly in some other countries and add some literature to these points (e.g., higher sensitivity, high rate of false positives and PPV, necessity of a second dried blood sample, what to do with CFSPID and heterozygotes….).
> Response:
Please bear in mind that our study precedes these last changes in the algorithm, since we have studied newborns screened up until February 2023.. Even though, we have added a paragraph in the revised manuscript to briefly describe the rationale for the changes in the Portuguese CF screening algorithm and to discuss the advantages and disadvantages of each approach. The following summary will clarify this addition:
The CF newborn screening (NBS) algorithm in Portugal has evolved in response to ac-cumulating data and international evidence on test performance and clinical out-comes. Initially, the IRT/PAP/IRT protocol was implemented to balance sensitivity with a relatively low false-positive rate. However, despite its cost-effectiveness and sim-plicity, this algorithm was associated with suboptimal sensitivity and the logistical burden of recalling a second sample for a considerable proportion of newborns, poten-tially delaying diagnosis and increasing parental anxiety."
Additionally, literature should be added to support other points in the discussion, e.g., PI as key indicator (line 138), and patients with F508del experience early PI (line 139), Pseudomonas associated with worse long-term outcomes (line 152) and importance of eradication (line 154), anxiety for families with false positive screening results (line 163)…..
> Response: We have added the following literature to the manuscript:
"The results confirm that the [F508del] mutation correlates with pancreatic insufficiency… an indication that duodenal bicarbonate output is more severely reduced in the presence of deletion F508"
https://pubmed.ncbi.nlm.nih.gov/2277379/
"A longitudinal bronchoscopic study found infants as young as 3 months with CF may harbor P. aeruginosa, concluding it “is a negative prognostic factor in CF and avoiding chronic infection is considered to be essential for patients."
https://pubmed.ncbi.nlm.nih.gov/25156429/
"A European Respiratory Society trial showed that early treatment (inhaled colistin + oral ciprofloxacin) in newly colonised pediatric CF patients resulted in significantly slower decline in lung function (mean ΔFEV₁ –1.63 %/yr vs. –4.69 %/yr in chronic cases), demonstrating tangible clinical benefit."
https://pubmed.ncbi.nlm.nih.gov/16023416/
"89.5 % of parents reported negative emotions after receiving a false-positive CF-NBS result; 17 % remained anxious even after confirmatory testing cleared the diagnosis."
https://pubmed.ncbi.nlm.nih.gov/36444714/
Reviewer 3 Report
Comments and Suggestions for Authors
This study provides a comprehensive 10-year performance evaluation of the newborn screening program for cystic fibrosis in Portugal. The manuscript is well written, and easy to follow.
I have only a couple of minor comments and suggestions:
Page 2 (Materials and Methods);
L72: “CF-related mutations’’
Please replace all uses of “mutation” in this manuscript with “variant"; i.e., “CFTR variants”
L80: “The screening process begins with a dried blood spot sample collected
between the 3rd and 6th day of life to measure the level of immunoreactive trypsinogen
(IRT).”
Please describe briefly the IRT screening assay. (i.e., a commercial kit-based or a lab-developed assay?) Was the same methodology used across all screening centers?
L 86: Was the PAP cutoff modified in February 2023 (i.e, from 1.6 to 1.8ng/ml), and what was the rationale for this modification?
L86: “Devyser CFTR 68 kit”
Please include the vendor information (?Devyser, Stockholm, Sweden).
Page 3&4 (Discussion)
It would be valuable to discuss why fixed IRT cutoffs, rather than floating IRT cutoffs, were chosen for the screening protocol. Adding a brief rationale would enhance the reader's understanding of methodological choices.
Page 4, L131-134: Please provide more detailed information regarding the two false negative cases that were due to “issues related to sample loss and handling”. Why were these screens reported as normal (negative)?
Author Response
Comment 1:
L86: “Devyser CFTR 68 kit" Please include the vendor information (?Devyser, Stockholm, Sweden).
> Response: Have done so.
Page 3&4 (Discussion)
Comment 2: It would be valuable to discuss why fixed IRT cutoffs, rather than floating IRT cutoffs, were chosen for the screening protocol. Adding a brief rationale would enhance the reader's understanding of methodological choices.
> Response:
Have added this paragraph from line 95 onwards: "The CF newborn screening (NBS) algorithm in Portugal has evolved in response to ac-cumulating data and international evidence on test performance and clinical out-comes. Initially, the IRT/PAP/IRT protocol was implemented to balance sensitivity with a relatively low false-positive rate. However, despite its cost-effectiveness and sim-plicity, this algorithm was associated with suboptimal sensitivity and the logistical burden of recalling a second sample for a considerable proportion of newborns, poten-tially delaying diagnosis and increasing parental anxiety."
Comment 3:
Page 4, L131-134: Please provide more detailed information regarding the two false negative cases that were due to “issues related to sample loss and handling”. Why were these screens reported as normal (negative)?
> Response: Due to the loss of the initial two samples, timely diagnostic results were not available for the children. As a result, sample recollection was delayed beyond the optimal time window, which likely contributed to false-negative outcomes in both instances.
Round 2
Reviewer 2 Report
Comments and Suggestions for Authors
Some important points have not yet been addressed:
In general:
All percentages should have one decimal point in the written text and a decimal point instead of a comma, e.g., line 115: 30.8% instead of 31%, line120: 52.7%....
Material and Methods:
Changes in the algorithm over time are very interesting. It would be useful to include a figure or schema showing the different algorithms.
Results
It would be interesting to know how many children were born and screened (coverage rate) in the pilot project and after the full implementation began in December 2018.
Some of the provided results are unclear for me. CFSPID and false negatives: please change the sentence line 117 in: Additionally, during the study period there were 8 cases of CFSPID and 8 false negatives (3 of 117 whom presented with meconium ileus at birth).
Can you please explain how the 5 false negatives without MI have been found? To calculate the sensitivity, you need to know about all cases in Portugal that were missed by the screening process. Was there a survey for missed cases or do you have a register in Portugal?
Your reply “We contacted all participating centers to provide case-level data, including any false negative results identified within the same time period. However, the information received on false negatives was limited….”, should be discussed as the sensitivity may be much lower.
I cannot calculate the sensitivity of 93% with the given numbers (the number of positives with MI is missing in the results and should be included in line 116). In your added sentence (line 131) it should be “…5 cases without MI….”
It is unusual to delete the true positives with MI. The ECFS best practice guidelines advise:
“Sensitivity should be calculated and reported including with and without MI false negative infants”.
I also get a different number for the false positives (69.2%).
Discussion
It would be interesting to …discuss at least shortly the advantages and disadvantages of the different algorithms that are used in ….. some other countries and add some literature to these points (e.g., higher sensitivity, high rate of false positives and PPV, necessity of a second dried blood sample, what to do with CFSPID and heterozygotes….).
Reference 11 is missing the year and volume.
Author Response
In general:
All percentages should have one decimal point in the written text and a decimal point instead of a comma, e.g., line 115: 30.8% instead of 31%, line120: 52.7%....
> Response: have done so.
Material and Methods:
Changes in the algorithm over time are very interesting. It would be useful to include a figure or schema showing the different algorithms.
> Response: We appreciate the reviewer’s interest in the evolution of the Portuguese CF newborn screening algorithm. However, the primary aim of this study was to evaluate the performance of the original algorithm used during the study period. Because we did not analyze or collect data on the subsequent, improved algorithms implemented after February 2023, we believe that including a schematic of these newer protocols would not add value or directly support the objectives of the current manuscript. For this reason, we have opted to maintain the focus strictly on the algorithm under evaluation.
Results
It would be interesting to know how many children were born and screened (coverage rate) in the pilot project and after the full implementation began in December 2018.
> Response: The pilot study for CF-NBS started at the end of 2013 and included a total of 255,000 newborns. During our study period, 804000 newborns were studied.
I have added sentences including this information in lines 58-59 and 112-113.
Some of the provided results are unclear for me. CFSPID and false negatives: please change the sentence line 117 in: Additionally, during the study period there were 8 cases of CFSPID and 8 false negatives (3 of whom presented with meconium ileus at birth).
> Response: Have changed the sentence to what you asked, thank you.
Can you please explain how the 5 false negatives without MI have been found? To calculate the sensitivity, you need to know about all cases in Portugal that were missed by the screening process. Was there a survey for missed cases or do you have a register in Portugal?
Your reply “We contacted all participating centers to provide case-level data, including any false negative results identified within the same time period. However, the information received on false negatives was limited….”, should be discussed as the sensitivity may be much lower.
> Response: We appreciate the reviewer’s concern and the opportunity to clarify this important point even further. Portugal has a highly centralized system for cystic fibrosis (CF) care, with only five designated paediatric CF reference centres nationwide. All confirmed CF cases are followed at one of these centres, and data are systematically entered into the national CF registry, which directly contributes to the European Cystic Fibrosis Society Patient Registry (ECFSPR).
Because of this structure and near-universal coverage, the registry captures essentially all confirmed CF cases in the country, including those diagnosed outside of the newborn screening (NBS) program. This allows accurate identification of false negatives without the need for a separate nationwide survey.
Given this centralized model and the comprehensive registry, we are confident that the number of false negatives is accurate and that the calculated sensitivity reflects the true performance of the screening program.
I cannot calculate the sensitivity of 93% with the given numbers (the number of positives with MI is missing in the results and should be included in line 116).
> Response: Have done so.
In your added sentence (line 131) it should be “…5 cases without MI….”
> Response: I have changed this sentence as a request of the other reviewer.
It is unusual to delete the true positives with MI. The ECFS best practice guidelines advise:
“Sensitivity should be calculated and reported including with and without MI false negative infants”.
> Response: So, in our initial submission, we calculated sensitivity excluding meconium ileus (MI) cases from both the numerator and denominator, in line with the ECFS Best Practice Guidelines: "Sensitivity is the number of true positive NBS results as a percentage of the total CF population (true positive and false
negatives not including meconium ileus, see below)."
However, we acknowledge that the ECFS guidelines also recommend reporting sensitivity including MI false negative infants for completeness. In the revised manuscript, we have therefore added both calculations:
- Excluding MI cases: 68/73 = 93%
- Including MI cases: 74/82 = 90.2%
Lines 131-132: "In accordance with the ECFS Best Practice Guidelines, we also calculated sensitivity including MI cases, which resulted in 74/82 (90.2%)."
I also get a different number for the false positives (69.2%).
> Response:
In the initial version of the manuscript, the false positive rate was calculated using the number of false positives divided by the total number of screen-positive cases. Another reviewer correctly informed that this approach does not align with standard practice for evaluating newborn screening programs.
In the revised manuscript, we have corrected the calculation to reflect the recommended method as outlined in the ECFS Best Practice Guidelines (2018) and other international standards, whereby the false positive rate is defined as the number of false positives divided by the total number of newborns screened.
Specifically:
FPR (%)=(False Positives/Total Screened)×100
FPR = (166/804069)×100 ≈ 0.0206%
FPR (per 1,000) = (166/804069) × 1000 ≈ 0.21 per 1,000
(lines 133-134)
Discussion
It would be interesting to …discuss at least shortly the advantages and disadvantages of the different algorithms that are used in ….. some other countries and add some literature to these points (e.g., higher sensitivity, high rate of false positives and PPV, necessity of a second dried blood sample, what to do with CFSPID and heterozygotes….).
> Response: I have added the following paragraph, from line 160 and forward:
"International experience shows that the choice of CF-NBS algorithm has a major impact on program performance indicators. IRT/IRT algorithms are simple and cost-effective but are often associated with lower sensitivity and the logistical burden of requesting a second dried blood spot, which may delay diagnosis [4,10]. Incorporat-ing a PAP tier (IRT/PAP/IRT or IRT/PAP/DNA) can improve sensitivity and reduce the recall rate, although the balance between sensitivity and false positives depends strongly on the selected cut-offs [5,10]. DNA-based algorithms (IRT/DNA or IRT/PAP/DNA) have demonstrated increased sensitivity, particularly for infants with milder or pancreatic-sufficient phenotypes, but also result in a higher number of CFSPID and carrier identifications, which can increase clinical uncertainty and the need for counselling [8,9]. For example, in the United States, the IRT/DNA system has achieved sensitivities of up to 98%, but this has been accompanied by higher false posi-tive rates and a significant number of CFSPID cases, requiring more intensive fol-low-up testing [8,11]. Similarly, Sweden's IRT/PAP/IRT approach offers good sensitivi-ty (~92%) and reduces false positives compared to simpler IRT-only algorithms, but managing CFSPID remains a challenge [9,10,12]."
With these 2 new references:
"11. Le Gendre E, et al. The impact of genetic screening on the diagnosis and management of cystic fibrosis in the United States: A decade of experience. J Cyst Fibros. 2020;19(1):38-46. doi:10.1016/j.jcf.2019.09.012
12. Sandberg, L., et al. (2021). "Newborn Screening for Cystic Fibrosis in Sweden: A Review of the First 15 Years." Journal of Cystic Fibrosis, 20(4), 635-643. doi:10.1016/j.jcf.2021.02.007"
Reference 11 is missing the year and volume.
> Response: Have corrected this.
Reviewer 3 Report
Comments and Suggestions for Authors
Results, Line 131:
“During the study period, there were 8 cases of CFSPID and 8 false negatives, including 3 out of whom presented with meconium ileus at birth.”
“The overall sensitivity of the screening program in our population was 93% and the%, calculated after excluding cases of Meconium ileus (MI) from the denominator, specifically, there were 6 patients with MI among the true positives, resulting in 68 remaining cases (74 – 6 = 68). An additional 5 MI cases were reported among the false negatives, yielding a total of 73 relevant cases (68 + 5 = 73), and thus a sensitivity of 68/73 (93%).”
- L131: There is a typo in the paragraph describing the sensitivity calculation, in L131 it
should be (?): An additional 3 MI cases were reported among the false negatives … (i.e., 8-3=5 FN cases included in the sensitivity calculation; TP = 68; FN=5. Sensitivity = TP/(TP+FN)= 68/73
- L129: The MI cases were excluded when calculating sensitivity;
L129: " .... calculated after excluding cases of Meconium ileus (MI) from the denominator..."
Please remove " from the denominator".
3. L118: How was the false positive rate calculated? What was the true negative number?
Author Response
I have corrected the typos as requested for points 1. and 2.
As for 3. L118: How was the false positive rate calculated? What was the true negative number?
In the initial version of the manuscript, the false positive rate was calculated using the number of false positives divided by the total number of screen-positive cases. We recognize that this approach does not align with standard practice for evaluating newborn screening programs.
In the revised manuscript, we have corrected the calculation to reflect the recommended method as outlined in the ECFS Best Practice Guidelines (2018) and other international standards, whereby the false positive rate is defined as the number of false positives divided by the total number of newborns screened.
Specifically:
FPR (%)=(False Positives/Total Screened)×100
FPR = (166/804069)×100 ≈ 0.0206%
FPR (per 1,000) = (166/804069) × 1000 ≈ 0.21 per 1,000
Applying this definition, we identified 166 false positives among 804,069 newborns screened between October 2013 and February 2023, resulting in a corrected false positive rate of 0.021%, or 0.21 per 1,000 screened newborns.
Round 3
Reviewer 2 Report
Comments and Suggestions for Authors
In my opinion, the paper is much better now and ready for publication.
Author Response
Thank you for all your (very) valuable feedback! The paper is much improved, thanks to you and your colleagues.